# Prevalence and Photobiology of Photosynthetic Dinoflagellate Endosymbionts in the Nudibranch *Berghia stephanieae*

**DOI:** 10.3390/ani11082200

**Published:** 2021-07-25

**Authors:** Ruben X. G. Silva, Paulo Cartaxana, Ricardo Calado

**Affiliations:** ECOMARE & Centre for Environmental and Marine Studies (CESAM), Department of Biology, University of Aveiro, 3810-193 Aveiro, Portugal; pcartaxana@ua.pt

**Keywords:** bleaching, *Exaiptasia diaphana*, pulse amplitude modulated (PAM) fluorometry, symbiosis, stenophagy

## Abstract

**Simple Summary:**

Some sea slugs have evolved highly specialized feeding habits and solely prey upon a reduced number of species. This is the case of *Berghia stephanieae*, a sea slug that feeds exclusively on glass anemones, such as *Exaiptasia diaphana*. Glass anemones host photosynthetic microalgae that *B. stephanieae* ingest when preying upon *E. diaphana*. The association between these photosynthetic microalgae and sea slugs appears to be limited in time, particularly if *B. stephanieae* is deprived of prey hosting these microalgae. In the present study, we validate the use of a non-invasive and non-destructive approach that allows monitoring the persistence of this association in live sea slugs by measuring chlorophyll fluorescence. A complete loss of photosynthetic microalgae was observed within 8 days when animals were deprived of food or fed glass anemones with no microalgae (bleached anemones). As such, the association between *B. stephanieae* and photosynthetic microalgae acquired when preying glass anemones is not a true symbiosis. Future studies may use the technique here described to monitor the prevalence of the association between sea slugs and photosynthetic microalgae, particularly under bleaching events that will impair sea slugs to acquire microalgae by preying upon their invertebrate hosts.

**Abstract:**

*Berghia stephanieae* is a stenophagous sea slug that preys upon glass anemones, such as *Exaiptasia diaphana*. Glass anemones host photosynthetic dinoflagellate endosymbionts that sea slugs ingest when consuming *E. diaphana*. However, the prevalence of these photosynthetic dinoflagellate endosymbionts in sea slugs appears to be short-lived, particularly if *B.*
*stephanieae* is deprived of prey that host these microalgae (e.g., during bleaching events impacting glass anemones). In the present study, we investigated this scenario, along with food deprivation, and validated the use of a non-invasive and non-destructive approach employing chlorophyll fluorescence as a proxy to monitor the persistence of the association between sea slugs and endosymbiotic photosynthetic dinoflagellates acquired through the consumption of glass anemones. *Berghia stephanieae* deprived of a trophic source hosting photosynthetic dinoflagellate endosymbionts (e.g., through food deprivation or by feeding on bleached *E. diaphana*) showed a rapid decrease in minimum fluorescence (F_o_) and photosynthetic efficiency (F_v_/F_m_) when compared to sea slugs fed with symbiotic anemones. A complete loss of endosymbionts was observed within 8 days, confirming that no true symbiotic association was established. The present work opens a new window of opportunity to rapidly monitor in vivo and over time the prevalence of associations between sea slugs and photosynthetic dinoflagellate endosymbionts, particularly during bleaching events that prevent sea slugs from incorporating new microalgae through trophic interactions.

## 1. Introduction

Mutualistic symbioses are characterized by specialization of each of the species taking part in these remarkable associations, as each one of them specializes in absorbing different nutrients, producing different metabolites, or providing an array of services that complement each other [1]. These associations commonly occur in the marine environment between different marine invertebrate taxa and photosynthetic dinoflagellate endosymbionts [2]. These highly diverse Symbiodiniaceae dinoflagellates are often still popularly termed “zooxanthellae,” although not all zooxanthellae are restricted to this taxa [3]. The most well-known associations described for these photosynthetic dinoflagellate endosymbionts are the ones established with tropical reef building corals, with up to 99% of the carbon fixed by these endosymbionts being translocated to its coral host as an energy source [4]. Some gastropod mollusks within order Nudibranchia also establish this type of association with Symbiodiniaceae dinoflagellates that they acquire through the ingestion of dietary prey [5,6]. The benefits associated with the retention of Symbiodiniaceae dinoflagellates by these sea slugs are yet to be completely understood, although a few studies have already elaborated on this subject [7,8,9]. One of the potential benefits suggested for this association is crypsis, the ability of a given animal to be less conspicuous in its natural environment, thus avoiding being detected by predators [5,10,11]. Another suggested benefit is the ability of sea slugs to endure periods of food shortage, thus being able to search the environment for suitable food sources [5,12].

The aeolid nudibranch *Berghia stephanieae* (Figure 1) is one such sea slugs that can retain photosynthetic dinoflagellate endosymbionts from its dietary prey. This nudibranch was first described as *Aeolidiella stephanieae* by Valdés [13]. In 2013, the species was assigned to genus *Berghia* by Carmona et al. [14], supported by molecular phylogenetic evidence using COI and 16S mitochondrial and H3 nuclear markers. Being a stenophagous species, as are several other nudibranchs, *B. stephanieae* feeds exclusively on glass anemones, such as *Exaiptasia diaphana* (Rapp, 1829) (Cnidaria: Actiniaria), which commonly host photosynthetic dinoflagellate endosymbionts [15]. This specialized feeding behavior has made this nudibranch species highly popular in the marine aquarium trade where it is erroneously traded as *Berghia verrucicornis* and fetches high retail values [16]. Indeed, this sea slug is popular amidst marine aquarium keepers for its ability to control glass anemone outbreaks in reef aquariums that may damage precious hard corals and other marine invertebrates commonly displayed [9,13]. This sea slug is also considered a suitable model organism for laboratory research, due to its hardiness, regular oviposition, short embryonic and larval period, short generation time and food availability [9], features that also make their captive production appealing for those farming marine ornamental species [17]. Furthermore, the ability of *B. stephanieae* to retain Symbiodiniaceae dinoflagellates from its prey makes this nudibranch one of the best models for ecological studies addressing mollusk–dinoflagellate endosymbiosis [8,9,18,19].

The symbiotic glass anemone *E. diaphana*, on which *B. stephanieae* predates, hosts Symbiodiniaceae dinoflagellates and, as such, is prone to be impacted by climatic events that promote bleaching (the loss of photosynthetic dinoflagellate endosymbionts) [20]. Consequently, this sea slug is also prone to be either directly or indirectly impacted by bleaching, due to its stenophagous diet [18]. Highly invasive techniques are commonly employed to study the impacts of the disruption of the association between nudibranchs and their photosynthetic dinoflagellate endosymbionts. Some of the most commonly employed techniques rely on histological and ultrastructural observations, which often leads to the sacrifice of the study subject [5,9]. However, Wägele and Johnsen [10] introduced an innovative approach to monitor photosynthesis in sea slugs [21,22,23,24]—the use of pulse amplitude modulated (PAM) fluorometry. This approach that measures chlorophyll *a* fluorescence had already been used to study other marine taxa, such as corals [25], seagrasses [26], macroalgae [27], sponges [28], and microphytobenthos [29,30], prior to being used on sea slugs. This method rapidly became popular due to the ease with which one can gather information on the state of photosystem II (PSII) over time, using a non-invasive and non-destructive approach [31].

The present work aimed to: (1) validate a non-invasive method to determine the abundance and photosynthetic efficiency of photosynthetic dinoflagellate endosymbionts in *B. stephanieae*; (2) evaluate the effect of different light and diets on the prevalence of photosynthetic dinoflagellate endosymbionts in *B. stephanieae*; and (3) compare the profiles of photosynthetic pigments present in *B. stephanieae* and *E. diaphana*.

## 2. Materials and Methods

### 2.1. Species Identification through DNA Barcoding

#### 2.1.1. DNA Extraction

One small adult specimen of *Berghia stephanieae* (~15 mm in total length), acquired from TMC Iberia (Lisbon, Portugal), and one specimen of *Exaiptasia diaphana* from our monoclonal culture, were preserved in ethanol (96%). Genomic DNA was extracted from these specimens using the DNeasy Blood and Tissue Kit (Qiagen) according to the manufacturer’s instructions.

#### 2.1.2. Polymerase Chain Reaction Amplification of 16S and COI Markers 

Amplification of partial regions of mitochondrial 16S rDNA (~500 bp) and cytochrome c oxidase subunit I (COI) (~700 bp) was carried out using the following primers: 16SAR (forward: 5′-CGCCTGTTTATCAAAAACAT-3′) and 16SBR (reverse: 5′-CCGGTCTGAACTCAGATCACGT-3′) for 16S rDNA [32] and LCO1490 (forward: 5′- GGTCAACAAATCATAAAGATATTGG-3′) and HCO2198 (reverse: 5′-TAAACTTCAGGGTGACCAAAAAATCA-3′) for COI [33]. PCR reaction mixtures performed used a final volume of 25 μL: 12.5 μL of DreamTaq PCR Master Mix, which contains Dream Taq DNA polymerase, optimized Dream Taq buffer, MgCl_2_ and dNTPs (Fermentas, Vilnius, Lithuania) (Fisher Scientific), 0.2 μM of each primer, and 1 μL of template DNA.

DNA amplification was performed with a Veriti 96-well Thermal Cycler (Applied Biosystems). For COI amplification there was an initial denaturation of 5 min at 95 °C, followed by 35 cycles of 1 min at 95 °C, denaturation stage, 45 s at 45 °C (primer-specific annealing temperature) and 1 min and 30 s at 72 °C, for extension. After these cycles, there was a final 5 min period of extension at 72 °C. Amplification of 16S was composed of an initial 5 min denaturation stage at 95 °C, followed by 40 thermal cycles of a 45 s denaturation stage at 95 °C, 45 s at 52 °C (primer-specific annealing temperature), and a 1 min extension stage at 72 °C. The cycles were followed by a final 7 min extension phase at 72 °C. In the case of *E. diaphana*, the primer-specific annealing temperature used for COI was 52 °C; for 16S, the extension stage used during the cycles was of 2 min instead of 1 min. DNA amplification was verified using electrophoresis on an agarose gel stained with GelRed (Biotium) and later visualized under UV light.

#### 2.1.3. DNA Sequencing and Analysis

PCR-amplified fragments from both markers (16S/COI) were sequenced on both orientations, with Sanger sequencing being performed by STAB VIDA (Portugal). MEGA version 6 was used to reach the consensus sequence, which was then blasted (Basic Local Alignment Search Tool, BLAST, National Center for Biotechnology Information, NCBI) on the GenBank database (http://www.ncbi.nlm.nih.gov/genbank/ (accessed on 6 November 2019)) to accurately identify each species.

### 2.2. Monoclonal Culture of Glass Anemones Exaiptasia diaphana

One specimen of *Exaiptasia diaphana*, originating from a live rock purchased from TMC Iberia (Lisbon, Portugal), was placed in one of two 260 L glass tanks (350 mm × 500 mm × 1500 mm) that were part of a 640 L life support system filled with artificial seawater, prepared using freshwater previously purified by reverse osmosis, using a V2 Pure 360 Reverse Osmosis System (Tropical Marine Centre, Hertfordshire, United Kingdom), and Red Sea Coral Pro Salt according to the manufacturer’s instructions. The anemone was allowed to asexually reproduce through pedal laceration of its basal peduncle, and after one month some monoclonal anemones were placed in the second 260 L tank to enhance their monoclonal propagation. All anemones were fed daily ad libitum with newly hatched *Artemia* sp. nauplii (Artemia Koral GmbH, Nürnberg, Germany) and kept under a 12 h light:12 h dark photoperiod using two Hailea sunshine T5 80 W fluorescent lamps delivering a photosynthetic active radiation (PAR) of 80 μmol photons m^−2^ s^−1^.

The 640 L life support system employed was equipped with a 120 L sump housing a Deltec SC 1455 internal protein skimmer, 0.006 m^3^ of 20 mm diameter bioballs, a TMC UV sterilizer P1 of 55 W, and two 300 W Eheim Jäger heaters. A cooling unit (Hailea 500AC chiller) was connected to an Eheim universal 1200 water pump placed inside the sump for temperature control. Water was returned to tanks with an Eheim universal 2400 pump, creating a flow of 380 L h^−1^ on each tank. Water temperature was kept stable at 26 ± 1 °C, as well as salinity that was kept stable at 35 ± 1 by using an osmoregulator connected to a water pump placed inside a reservoir with freshwater purified by reverse osmosis to compensate for water evaporation. To ensure water quality, 70 L water changes were performed twice a week. Ammonia and nitrites were always undetectable, with nitrates being recorded below 0.10 ppm, with these parameters being checked biweekly using Salifert (Duiven, Holland) colorimetric tests.

### 2.3. Captive Culture of Sea Slugs Berghia stephanieae

Thirty captive cultured small adults of *B. stephanieae* (~15 mm in total length) were acquired from TMC Iberia (Lisbon, Portugal) and housed in a similar system to that described above for glass anemones, with the sole difference being that one of the two 260 L (350 mm × 500 mm × 1500 mm) glass tanks was used as a water bath and housed four 21 L (380 mm × 200 mm × 280 mm) plastic boxes with a sandy bottom composed of a layer of 1 mm sand, followed by a layer of 2 mm coral gravel and a layer of 10 mm oyster shell fragments. Each plastic box was provided with a clay pot to serve as a hideout for *B. stephanieae*. Each of the above-mentioned boxes had an individual water inflow set at 12 L h^−1^. Inflowing water was forced to pass through the sandy bottom of the plastic boxes, to which 1 mm mesh screens had been previously fitted so water could outflow and, as such, no additional mesh screens had to be employed to prevent sea slugs from escaping the plastic boxes. Two of these boxes were used to stock 15 small adults each, while the other two boxes were left empty to receive juvenile specimens born in the laboratory (F1). 

All specimens of *B. stephanieae* were kept under a 10 h light:14 h dark photoperiod provided by an illumination system identical to that described above for glass anemones and fed daily ad libitum with monoclonal *E. diaphana*. As recommended in the literature, adults were fed with larger-sized anemones (5 to 8 mm of pedal disk diameter), while juveniles were supplied with smaller-sized specimens (2 to 4 mm of pedal disk diameter) [13]. All experiments detailed below employed only specimens cultured in the laboratory (F1) from the broodstock initially acquired.

### 2.4. Relation between the Abundance of Photosynthetic Dinoflagellate Endosymbionts Hosted by Berghia stephanieae and F_o_

To establish a non-invasive method to infer the abundance of photosynthetic dinoflagellate endosymbionts in *B. stephanieae*, 10 nudibranchs were paired and distributed over five 60 L (300 mm × 400 mm × 500 mm) aquariums, fitted with an Eheim thermocontrol 50 heater (50 W) and an Eheim Liberty 75 trickle filter. These aquariums were filled with synthetic seawater prepared as already described above.

Chlorophyll fluorescence measurements and symbiont counting were performed in the cerata of nudibranchs fed with monoclonal glass anemones, and after this, starved for a period of 5 and 10 days. Before each chlorophyll fluorescence measurement, each sea slug was dark-adapted for 45 min, after which ceras were collected from each animal and F_o_ was measured. Please see the section below detailing how chlorophyll fluorescence measurements were performed. The ceras were then smeared between a slide and a cover glass and multiple pictures were taken to cover the whole area of the cover glass. All pictures were taken with a Leica DM2500 light microscope (Leica Microsystems GmbH, Wetzalar, Germany) coupled to a Leica ICC50 W camera (Leica Microsystems GmbH, Germany). Images were then analyzed with the help of ICY bioimage analysis software version 2.1.0.1 (as described by de Chaumont et al. [34]) and ImageJ software version 1.52a.

### 2.5. Effect of Light Intensity and Starvation on the Abundance and Photosynthetic Efficiency of Photosynthetic Dinoflagellate Endosymbionts Hosted by Berghia stephanieae

To investigate the effect of light intensity and feeding regimes of host nudibranchs on the photobiology of photosynthetic dinoflagellate endosymbionts present in *B. stephanieae*, 20 sea slugs were randomly distributed under four experimental treatments combining two light intensities and two feeding regimes. Two light intensities were tested: 80 μmol photon m^−2^ s^−1^ (high light—HL), as employed by Monteiro et al. [9], and 10 μmol photon m^−2^ s^−1^ (low light—LL). Both high and low light were provided by Hailea sunshine T5 80 W fluorescent lamps, under a 10 h light:14 h dark photoperiod, by adjusting the distance of the light sources to the aquariums. The two different feeding regimes tested were the supply of *E. diaphana* from monoclonal culture (see above) hosting photosynthetic dinoflagellate endosymbionts and food deprivation (starved specimens). *Berghia stephanieae* fed under high light were considered control organisms.

Sea slugs were kept individually in two customized PVC trays described by Carvalho and Calado [35], with only the top portion of the tray being employed. This top portion of the tray was used floating in the same system employed to culture sea slugs, thus avoiding any unnecessary stress. Feeding was performed after chlorophyll fluorescence measurements by placing two medium-size anemones in containers stocked with fed sea slugs. Chlorophyll fluorescence measurements were performed as detailed below, every other day, starting at day zero and ending on the 14th day, at approximately midterm of daytime photoperiod.

### 2.6. Effect of Different Diets on the Abundance and Photosynthetic Efficiency of Photosynthetic Dinoflagellate Endosymbionts Hosted by Berghia stephanieae

To further evaluate the impact of different diets of *B. stephanieae* on the photobiology of their photosynthetic dinoflagellate endosymbionts, a new experiment was performed using 30 sea slugs; 10 were fed with symbiotic anemones (Fed Symb), 10 fed with bleached anemones (Fed Blea), and 10 were starved (Starved).

Symbiotic monoclonal glass anemones were produced as detailed above, with bleached anemones being produced by placing *E. diaphana* in two 360 L (300 mm × 1000 mm × 1200 mm) PVC boxes covered with an opaque lid, to keep them in darkness for three months while being fed daily ad libitum with newly hatched *Artemia* sp. nauplii (Artemia Koral GmbH, Germany). Each box was filled with natural seawater and equipped with an oyster shell biofilter, and an airlift system powered by an air pump to promote water circulation. A 50 L water change was performed weekly to safeguard water quality.

This experiment was also performed using the tray system already described above to stock sea slugs individually. Light was provided by two Hailea sunshine T5 80 W fluorescent lamps (80 μmol photon m^−2^ s^−1^). Fed sea slugs were supplied every other day either with two large-sized symbiotic anemones (5 to 8 mm of pedal disk diameter) or two large-sized bleached anemones after performing each chlorophyll fluorescence measurement (also every other day). These measurements ended on the 14th day and were always performed at approximately midterm of daytime photoperiod.

### 2.7. Effect of Different Diets on the Photosynthetic Pigment Profile of Photosynthetic Dinoflagellate Endosymbionts Hosted by B. stephanieae and E. diaphana

To ensure the success of the bleaching process and study the effect of different diets on the photosynthetic pigment profile of photosynthetic dinoflagellate endosymbionts hosted by *B. stephanieae* and *E. diaphana*, ten sea slugs were divided by two 60 L glass aquariums identical to those already described above. Five of these sea slugs were only fed with bleached anemones, while the other five were solely fed with symbiotic anemones for a period of 30 days. These animals were then collected and flash-frozen in liquid nitrogen and stored at −80 °C until further analysis. Five symbiotic and five bleached anemones were also collected and stored as previously described.

Pigment extraction and analysis by means of high-performance liquid chromatography (HPLC) was then performed on the collected samples, as described below, after lyophilization with a Unicryo MC4L −60 °C freeze dryer.

### 2.8. Chlorophyll Fluorescence Measurements

Chlorophyll fluorescence measurements were performed using a chlorophyll fluorometer Imaging-PAM, M-series, Mini-version (Walz, Germany), equipped with an IMAG-K7 camera. Prior to each measurement, sea slugs were dark-adapted for 45 min. Specimens were then placed, one at a time, on a watch glass with a minimal amount of water to restrain their movement, and the minimum fluorescence of dark-adapted samples (F_o_) was recorded. The parameters of the chlorophyll fluorometer were kept unchanged between measurements (measuring light intensity—2; frequency—1; gain—1; damping—1), so that different values of F_o_ being recorded could be reliably compared. All settings employed were previously tested to ensure the use of the available color range without risks of signal saturation. All animals were then returned to the dark for half an hour, placed on the same watch glass, and subjected to a saturating light to record the maximum quantum yield of photosystem II (F_v_/F_m_). F_v_/F_m_ was determined by calculating (F_m_ − F_o_)/F_m_, with F_m_ and F_o_ referring to the maximum and minimum fluorescence recorded on dark-adapted samples, respectively [31]. This procedure was performed separately from F_o_ to enable using different PAM settings. Numerical values of variable Chl fluorescence parameters were extracted from digital images using the imaging system software (Imaging Win, Heinz Walz GmbH, Effeltrich, Germany) by outlining sea slugs using the polygon tool.

### 2.9. Pigment Extraction and Analysis

Pigment extraction was performed following a modified procedure described by Mendes et al. [36]. Briefly, freeze-dried samples were macerated with a cold extraction solution (95% methanol buffered with 2% ammonium acetate) and then sonicated in a VWR Ultrasonic cleaner USC-T for 30 s. The samples were subsequently stored in the dark at −20 °C for 20 min. Extracts were filtered with PTFE 0.2 μm membrane filters and immediately injected into the HPLC. A Shimadzu HPLC, Prominence-I LC 2030C 3D Plus (Shimadzu Corp., Kyoto, Japan), was used to analyze extracted pigments. Chromatographic separation was carried out using a Supelcosil C18 column (250 mm length; 4.6 mm diameter; 5 μm particles; Sigma-Aldrich, USA) for reverse phase chromatography and a 35 min elution program. Photosynthetic pigments were identified by comparing absorbance peaks and retention times with the available literature [37]. Concentrations of photosynthetic pigments were calculated by using the signals recorded in the photodiode array detector and calibration curves constructed with pure crystalline standards from DHI (Hørsolm, Denmark).

### 2.10. Statistical Analysis

A linear regression was used to model the relationship between the concentration of photosynthetic dinoflagellate endosymbionts and F_o_ [38]. The existence of significant differences in the abundance and photosynthetic efficiency of photosynthetic dinoflagellate endosymbionts hosted by *B. stephanieae* under different light intensities and enduring starvation was investigated using the non-parametric Wilcoxon’s test, while the effects of different diets on the abundance and photosynthetic efficiency of *B. stephanieae* photosynthetic dinoflagellate endosymbionts were determined by using a Kruskal–Wallis test (paired with a post hoc Mann–Whitney test for pairwise comparisons) [38]. Significant differences in the concentrations of photosynthetic pigments of dinoflagellate endosymbionts hosted by *B. stephanieae* and *E. diaphana* were evaluated using a non-parametric Mann–Whitney U test [38]. The selection of non-parametric tests to evaluate the existence of significant differences was due to the violation of assumptions (e.g., data normality) required to employ parametric tests, even after performing a log(X + 1) data transformation. All statistical analyses were performed using the software IBM SPSS Statistics 27 (IBM Corp., Armonk, NY, USA). All graphics presented were created using the software SigmaPlot 11.0.

## 3. Results

### 3.1. Species Identification through DNA Barcoding

A BLAST analysis of the sea slug’s 16S and COI consensus sequence against the NCBI GenBank database confirmed the identification of the sea slug as a *B. stephanieae*. Both consensus sequence queries returned a result with a high query cover (16S—98%, COI—96%), both with an E value of 0.0 and a high percentage of identity (16S—99.75%, COI—99.70%).

BLAST of the anemone’s COI consensus sequence against the NCBI GenBank database resulted in the identification of the specimen surveyed as *Aiptasia pulchella*, a taxonomic identification currently unaccepted for *E. diaphana*. This query returned a result with a high query cover (99%), an E value of 0.0, and a high percentage of identity (99.56%).

### 3.2. Relation between the Abundance of Photosynthetic Dinoflagellate Endosymbionts Hosted by Berghia stephanieae and F_o_

F_o_ decreased when *B. stephanieae* endured starvation, while conspecifics fed with symbiotic anemones displayed a stable F_o_ (Figure 2).

The linear regression modeled to predict the concentration of photosynthetic dinoflagellate endosymbionts in the cerata of sea slugs using F_o_ as a proxy revealed that these two variables are significantly and positively correlated (F (1, 28) = 33.899, *p* < 0.001, R = 0.740) (Figure 3). The concentration of photosynthetic dinoflagellate endosymbionts can be predicted using the following equation:(1)Nsymbiontmm of cerata−1=15,662×(Fo)−758.73

### 3.3. Effect of Light Intensity and Starvation on the Abundance and Photosynthetic Efficiency of Photosynthetic Dinoflagellate Endosymbionts Hosted by Berghia stephanieae

Mortality occurred throughout the whole experiment, with starved specimens displaying 30% mortality and fed ones only 10% mortality. Starvation had a statistically significant effect on the minimum chlorophyll fluorescence of the sea slug’s photosynthetic dinoflagellate endosymbionts (Z = −2.366, *p* = 0.018), with the fed specimens displaying an average F_o_ of 0.285 at the end of the experimental period, while the starved conspecifics displayed an average F_o_ of 0.022 (Figure 4).

As for maximum quantum yield of PSII, starvation also had a statistically significant effect on F_v_/F_m_ of the sea slug’s photosynthetic dinoflagellate endosymbionts (Z = −2.201, *p* = 0.028), with the fed specimens maintaining an average F_v_/F_m_ of 0.6 throughout the experiment, while the starved conspecifics displayed a marked reduction of F_v_/F_m_ reaching 0 after 8–14 days (Figure 5). Irradiance levels tested showed no significant effect on the abundance (Z = −1.355, *p* = 0.176) and photosynthetic efficiency (Z = −1.826, *p* = 0.068) of photosynthetic dinoflagellate endosymbionts hosted by sea slugs.

### 3.4. Effect of Different Diets on the Abundance and Photosynthetic Efficiency of Photosynthetic Dinoflagellate Endosymbionts Hosted by Berghia stephanieae

Mortality was recorded in all treatments, with the starved specimens showing the highest mortality (at 60%), the fed specimens showing the lowest value (at 10%), and the individuals fed with bleached anemones showing a mortality of 20%. Statistically significant differences were found for the effect of different feeding regimes on the minimum chlorophyll fluorescence after 14 days (H (2) = 15.012, *p* = 0.001). A post hoc Mann–Whitney test revealed that individuals fed with symbiotic anemones presented a significantly higher F_o_ at 0.218 than conspecifics fed with bleached anemones at 0.018, U(n Fed Symb = 9, n Fed Blea = 8) = 0.000, Z = −3.475, *p* = 0.001, and starved ones at 0.022, U(n Fed Symb = 9, n Fed Blea = 4) = 0.000, Z = −2.777, *p* = 0.005 (Figure 6). Different diets also promoted a significant impact on maximum quantum yield of PSII of photosynthetic dinoflagellate endosymbionts hosted by *B. stephanieae* (H (2) = 15.202, *p* < 0.001) after 14 days. A post hoc Mann–-Whitney test showed that sea slugs fed with symbiotic anemones displayed a significantly higher F_v_/F_m_ at 0.473 than those fed with bleached anemones at 0.043, U(n Fed Symb = 9, n Fed Blea = 8) = 0.000, Z = −3.486, *p* < 0.001, and starved ones at 0.090, U(n Fed Symb = 9, n Fed Blea = 4) = 0.000, Z = −2.781, *p* = 0.005 (Figure 7).

### 3.5. Effects of Different Diets on the Photosynthetic Pigment Profile of Photosynthetic Dinoflagellate Endosymbionts Hosted by Berghia stephanieae and Exaiptasia diaphana

Bleaching was successfully achieved, as no photosynthetic pigments were found in *B. stephanieae* fed with bleached anemones. Furthermore, only two of the five bleached anemones tested presented measurable photosynthetic pigments, but all at very low concentrations (peridinin—0.03 mg g^−1^; chlorophyll *a*—0.01 mg g^−1^). Photosynthetic pigment profiles recorded for both *B. stephanieae* and *E. diaphana* are detailed in Table 1. Statistically significant differences were recorded in chlorophyll *c*_2_ (*p* = 0.027), peridinin (*p* = 0.016), diadinoxanthin (*p* = 0.028), and chlorophyll *a* (*p* = 0.016) concentrations between *B. stephanieae* and *E. diaphana*, with glass anemones having a higher concentration of photosynthetic pigments than sea slugs (Figure 8).

## 4. Discussion

The association between *Berghia stephanieae* and photosynthetic dinoflagellate endosymbionts was first studied in 1991 by Kempf [39] and was classified as a primitive mutualistic symbiosis. Since then, it has been the focus of additional studies that employed rather intrusive methods, including the removal of cerata and the eventual sacrifice of studied specimens to further elucidate this association [9,18,19]. Data retrieved in the present work employing a non-invasive and non-destructive approach allowed us to reliably infer over time the abundance of photosynthetic dinoflagellate endosymbionts within specimens of *B. stephanieae* exposed to different trophic conditions. By using the photosynthetic parameter F_o_ as a proxy for chlorophyll *a* within *B. stephanieae*, it is possible to estimate the abundance of photosynthetic dinoflagellate endosymbionts without having to sacrifice the specimens being surveyed.

In every experimental treatment performed where sea slugs were deprived of acquiring new photosynthetic dinoflagellate endosymbionts, either by starvation or by being fed bleached anemones, the values of F_o_ were sharply reduced with a complete loss of endosymbionts within eight days. This finding confirms that sea slugs lost their photosynthetic dinoflagellate endosymbionts over a short time and is in line with the findings of Monteiro et al. [9]. While employing an invasive approach, the previous authors considered that the association between *B. stephanieae* and Symbiodiniaceae did not meet the biological criteria to be termed as a mutualistic relationship, such as metabolite exchange and long-term retention [9,40,41]. Indeed, the retention time of photosynthetic dinoflagellate endosymbionts by *B. stephanieae* is rather short, with this sea slug being unable to sequester free-living Symbiodiniaceae and its somatic growth being rather unaffected by the presence or absence of dinoflagellates [9,39]. The concomitant decrease of maximum quantum yields (F_v_/F_m_) observed in sea slugs deprived of acquiring new photosynthetic dinoflagellate endosymbionts indicates the declining photosynthetic capacity of endosymbionts in *B. stephanieae*.

A study addressing different sea slug species known to host photosynthetic dinoflagellate endosymbionts recorded the presence of deteriorated cells in the fecal matter of these mollusks, thus suggesting that these sea slug species may obtain some nutritional benefits from digesting Symbiodiniaceae when deprived of food [6]. Our data showed no statistical difference when comparing the evolution of F_o_ of *B. stephanieae* enduring starvation or being fed with bleached anemones. This finding, alongside the fact that Kempf [39] found no evidence of active digestion of Symbiodiniaceae by *B. stephanieae*, indicates that the loss of photosynthetic dinoflagellate endosymbionts recorded is driven by the inability to selectively retain them if the sea slug is incapable of renewing the pool of Symbiodiniaceae through a trophic pathway. In other words, regular feeding on glass anemones hosting photosynthetic dinoflagellate endosymbionts is required to replace the losses that occur through defecation [39]. This dependence was experimentally supported in our study by sea slugs that were supplied symbiotic glass anemones ad libitum, whose photosynthetic dinoflagellate endosymbionts displayed relatively stable values of F_o_ and F_v_/F_m_ throughout the whole experimental period.

While the use of chlorophyll fluorescence is certainly worth exploring to gain new insights on the study of mollusk–photosynthetic dinoflagellate endosymbiont associations, one needs to consider that sea slugs are motile organisms, and their movement may significantly bias data being recorded. Every slight movement of the sea slug being monitored will induce non-physiological changes in the fluorescence signal being emitted by the photosynthetic dinoflagellate endosymbionts they host, which will ultimately compromise fluorescence-related outputs, such as steady-state light-response curves, rapid light-response curves, and F_v_/F_m_ [24,42].

While bleached anemones were kept in the dark for a long period of time (three months), they still appeared to harbor a small number of photosynthetic dinoflagellate endosymbionts, as residual levels of Chl *c*_2_, Peri, Diadino, and Chl *a* were recorded. This may be due to the prevalence of a very reduced number of photosynthetic dinoflagellate endosymbionts in glass anemones kept in darkness, as already suggested by Leal et al. [18]. Monteiro et al. [9] later confirmed this prevalence by directly counting photosynthetic dinoflagellate endosymbionts within bleached glass anemones, although these solely accounted for 0.5% of their initial number prior to a prolonged exposure to complete darkness. Nevertheless, the potential presence of this residual number of photosynthetic dinoflagellate endosymbionts in bleached anemones was not enough to allow detecting measurable photosynthetic pigments in *B. stephanieae* fed with bleached glass anemones.

## 5. Conclusions

Our study supports the finding of Monteiro et al. [9] that no true symbiotic association exists between *B. stephanieae* and the photosynthetic dinoflagellate endosymbionts that it acquires through the ingestion of *E. diaphana*. The use of chlorophyll fluorescence was successfully validated as a non-destructive approach that allows to rapidly monitor, in real time and in vivo, the persistence of photosynthetic dinoflagellate endosymbionts in sea slugs that predate marine invertebrates hosting these microalgae. This methodological approach can also be used to further investigate whether some groups of sea slugs have evolved mechanisms that enable them to retain photosynthetic dinoflagellate endosymbionts acquired through trophic interactions for longer periods. Such an adaptation may allow sea slugs to better endure the impact of bleaching events that strip photosynthetic marine invertebrates from photosynthetic dinoflagellate endosymbionts—that is, whether they contribute photosynthates relevant for the nutrition of sea slugs. Additionally, as Symbiodiniaceae is a remarkably diverse group, not only taxonomically but also functionally [3], and symbiont type can influence the trophic plasticity of glass anemones [43,44], it will certainly be worth investigating whether or not similar findings to the ones here described are recorded for *B. stephanieae* feeding on *E. diaphana* hosting different genera of Symbiodiniaceae.

## Figures and Tables

**Figure 1 animals-11-02200-f001:**
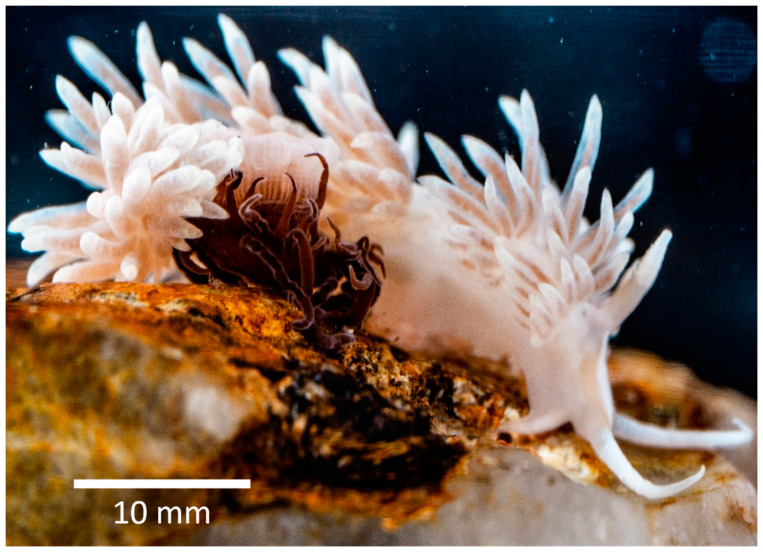
Close-up of *Berghia stephanieae* alongside its prey *Exaiptasia diaphana*.

**Figure 2 animals-11-02200-f002:**
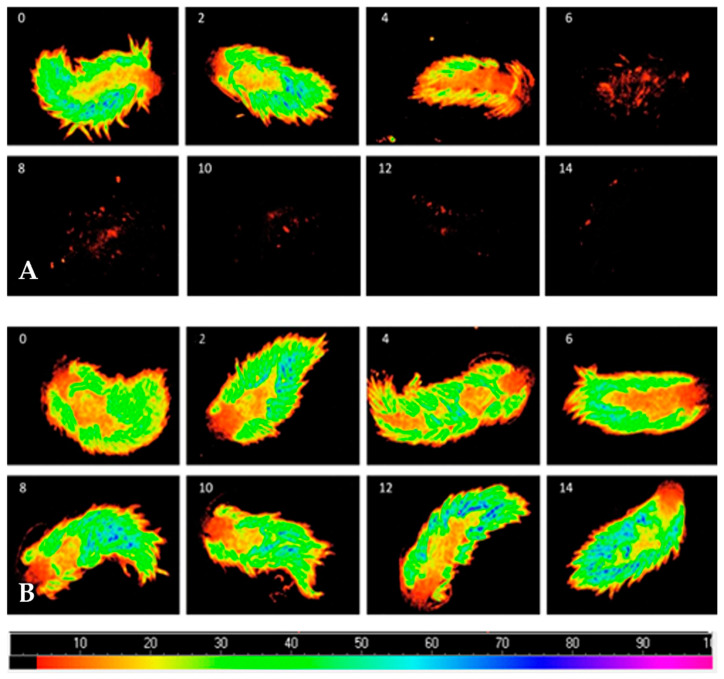
Evolution of the fluorescence parameter F_o_ monitored using an Imaging-PAM fluorometer in *Berghia stephanieae* enduring starvation (**A**) or being fed with symbiotic anemones (*Exaiptasia diaphana*) for 14 days (**B**). Numbers in the top left corner of each image refer to the number of days under each feeding treatment.

**Figure 3 animals-11-02200-f003:**
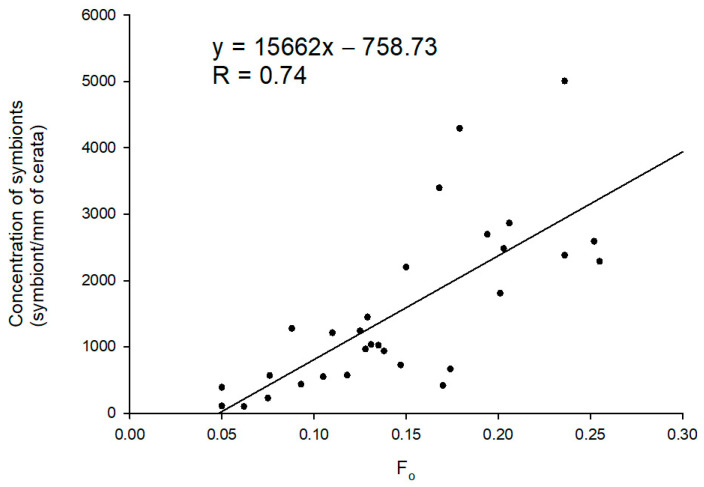
Correlation between the number of photosynthetic dinoflagellate endosymbionts within the cerata of *Berghia stephanieae* and fluorescence parameter F_o_ (n = 30).

**Figure 4 animals-11-02200-f004:**
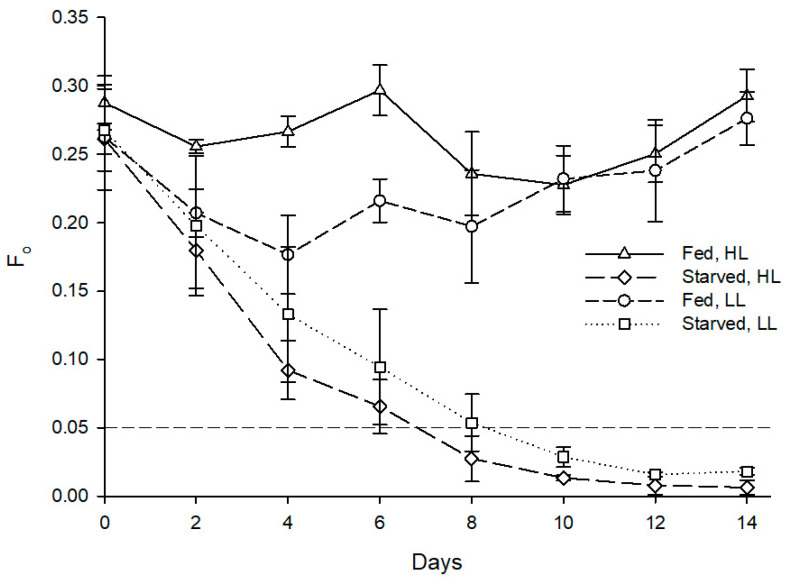
Evolution of fluorescence parameter F_o_ in photosynthetic dinoflagellate endosymbionts hosted by *Berghia stephanieae* reared under different light intensities and diets (mean ± std. error, starting n = 5). HL—high light (80 μmol photon m^−2^ s^−1^), LL—low light (10 μmol photon m^−2^ s^−1^). The dashed line indicates the threshold of F_o_ corresponding to the absence of endosymbionts detectable in sea slugs’ cerata using microscopy.

**Figure 5 animals-11-02200-f005:**
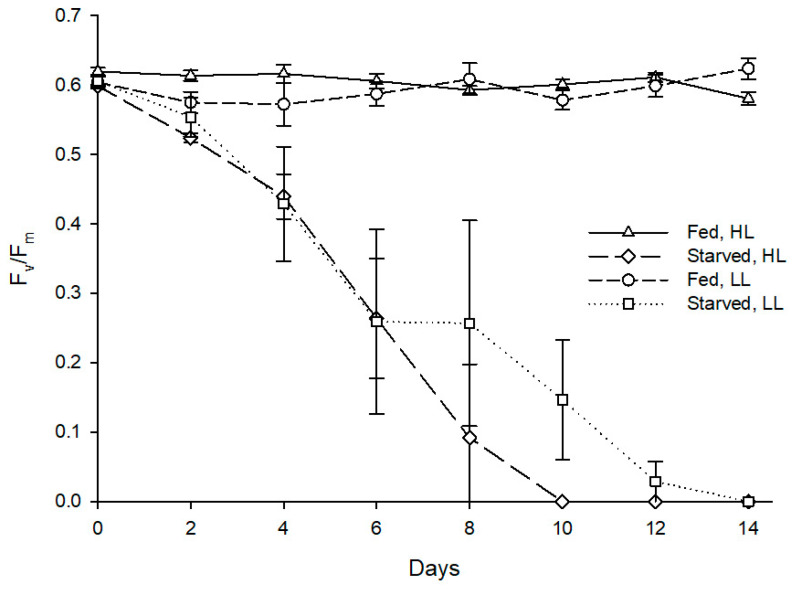
Evolution of fluorescence parameter F_v_/F_m_ in photosynthetic dinoflagellate endosymbionts hosted by *Berghia stephanieae* reared under different light intensities and diets (mean ± std. error, starting n = 5). HL—high light (80 μmol photon m^−2^ s^−1^), LL—low light (10 μmol photon m^−2^ s^−1^).

**Figure 6 animals-11-02200-f006:**
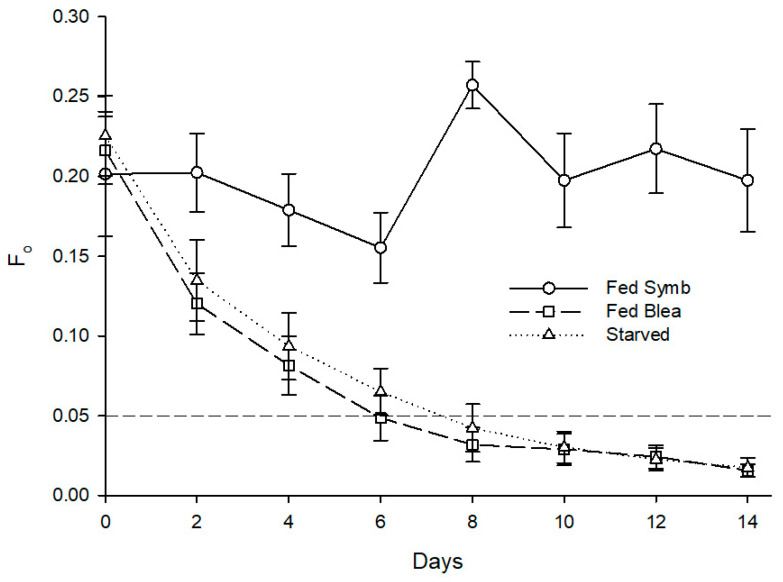
Evolution of fluorescence parameter F_o_ in photosynthetic dinoflagellate endosymbionts hosted by *Berghia stephanieae* reared under different diets (mean ± std. error, starting n = 10). Fed Symb—fed with symbiotic anemones (*Exaiptasia diaphana*), Fed Blea—fed with bleached anemones. The dashed line indicates the threshold of F_o_ corresponding to the absence of endosymbionts detectable in sea slugs’ cerata using microscopy.

**Figure 7 animals-11-02200-f007:**
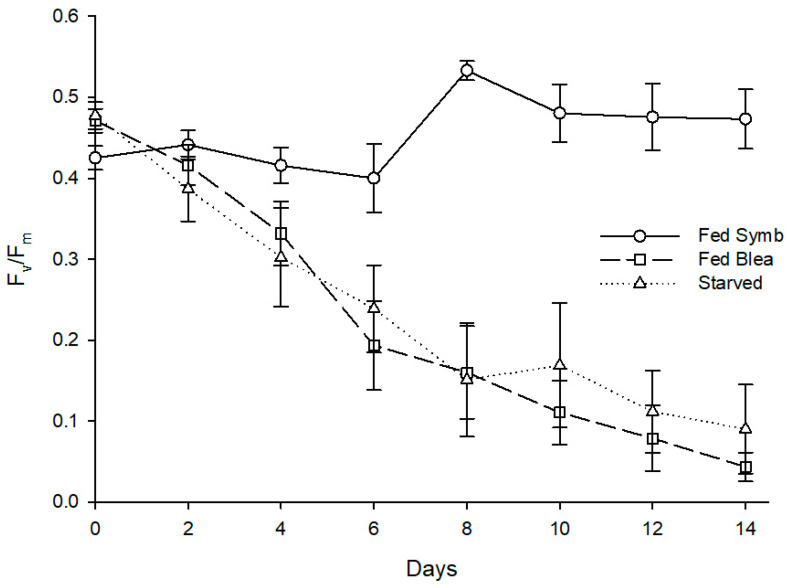
Evolution of fluorescence parameter F_v_/F_m_ in photosynthetic dinoflagellate endosymbionts hosted by *Berghia stephanieae* when exposed to different diets (mean ± std. error, starting n = 10). Fed Symb—fed with symbiotic anemones (*Exaiptasia diaphana*), Fed Blea—fed with bleached anemones.

**Figure 8 animals-11-02200-f008:**
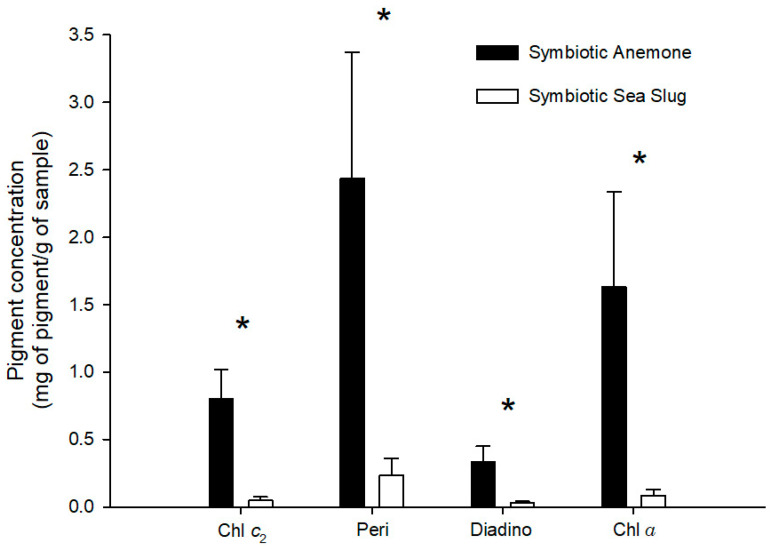
Concentration of photosynthetic pigments (mean ± std. error, n = 5) recorded in symbiotic anemones *Exaiptasia diaphana* and sea slugs *Berghia stephanieae* fed with symbiotic anemones (*Exaiptasia diaphana*). Chl *c*_2_—chlorophyll *c*_2_, Peri—peridinin, Diadino—diadinoxanthin, Chl *a*—chlorophyll *a*. An asterisk above a pigment indicates significant differences (*p* < 0.05).

**Table 1 animals-11-02200-t001:** List of photosynthetic pigments recorded in *Berghia stephanieae* and *Exaiptasia diaphana* with retention times and absorption maxima.

Photosynthetic Pigment (Abbreviation)	Retention Time (min)	Absorption Maxima (nm)
Peridininol	6.783	472
Chlorophyll *c*_2_ (Chl *c*_2_)	8.301	443, 580, 629
Peridinin (Peri)	10.521	476
Dinoxanthin (Dino)	14.264	418, 442, 472
Diadinoxanthin (Diadino)	14.764	420, 448, 478
Diadinochrome (Diadchr)	15.417	408, 430, 478
Chlorophyll *a* (Chl *a*)	24.258	430, 616, 663
Phaeophytin *a* (Phe *a*)	26.888	505, 607, 665

## Data Availability

All raw data are available as Appendix A.

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
