# Peer review of "Prevalence and Photobiology of Photosynthetic Dinoflagellate Endosymbionts in the Nudibranch Berghia stephanieae"

_animals, 2021, doi:10.3390/ani11082200_

Round 1

Reviewer 1 Report

This is a nicely composed research paper on a rather under-researched topic. Because it is true that PAM fluorometry has been used mainly on free-living plants/algae/cyanobacteria, and despite my below critical remarks, I do encourage its publication ESPECIALLY in an animal-related journal so as to further its use in research on animal-photosynthesiser symbioses. However, before publication in Animals, I ask the authors to consider the below points before submitting a final ms.

  1. The authors conclude correctly that the association between the sea slug and Symbiodinium, via glass anemones, is not a true symbiosis (since the algae disappear after a week of non-feeding or feeding on non-algal-containing anemones). However, I wonder if this is a novel finding? Are there ANY such symbioses in slugs that ARE “true”? If so, a) name them through citations and discuss them with relation to the present findings; b) If not, please downplay the finding of non-true symbiosis found here (by adding “… like in other slugs (refs.)“ or the like (e.g. on line 474 and/or earlier).
  2. Can the authors please relate to a question regarding the potential photosynthetic activity (Fv/Fm) stemmed from intact Symbiodinium (as indicated e.g. on lines 186 and 425) or from their chloroplasts once the rest of the dinoflagellates’ cells were digested? The latter is true for other slugs feeding on e.g. green macroalgae (called cleptoplasty or something like that), and is discussed for these animals by Wägele & Johnsen (2001, cited reference no. [6]). This should be addressed as an addition to the paper. [What about Aplysia – I seem to remember reading papers about that genus and its symbiotic chloroplasts lasting limited times… If relevant, please include it here.]
  3. Although much less than in free-living photosynthesisers, the use of PAM fluorometry on marine photosymbiotic animals has been established much more than inferred by the authors. Conspicuously missing are sponges (e.g. Ilan and Beer 1999, Coral Reefs 18: 74), and while corals were mentioned (reference [23]), the citation is ill chosen since they have been used more widely and much earlier (see e.g. Winters et al. 2003, Limnol. Oceanogr. 48: 1388 and other works by Winters and his group). Similarly, Cruz et al. 2015 is cited for using PAM fluorometry for macroalgae (line 90), but in fact they used it mainly for sea slugs. A better, earlier and more basic, reference would be e.g. Beer et al. 2000, Eur. J. Phycol. 35: 69-74 or other works by Beer and his group. So, I ask for a more balanced report on previous uses of PAM fluorometry in marine photosynthetic organisms.
  4. The authors praise the use of PAM fluorometry as a way not having to sacrifice the animals. Yes, while this method is non-intrusive, many animals were sacrificed in the preset research… 10-60% of the slugs died during the experiments, either by starvation or by being kept under non-ideal and unnatural conditions. While this was mentioned in passing (lines 339 and 368), this mortality was never discussed. Could it be that the results were affected by the mortality of even non-starved animals because they were kept under non-natural conditions? Please discuss this important matter.
  5. The authors claim that the Fo value proxied for “endosymbiont abundance” (e.g. lines 425-426) while, in truth, Fo reflects only the overall chlorophyll concentration. So, while a correlation certainly exists between chlorophyll concentration (e.g. in Fig. 8) and Fo, may it not be that the in vivo “endosymbiont abundance” declined less with time than claimed partly because there were less, or partly bleached, chloroplasts resulting in less chlorophyll per zooxanthellar cell? This is indicated also by the decrease in the maximal (potential) quantum yield (Fv/Fm) with time after ingesting the zooxanthellae, which could be a possible result of deficient photosynthesis based paralleled by the decreased pigments per cell. In addition, the scatter and, thus, low R value in Fig. 3 indicates that there may be another, additional, reason to the “concentration of symbionts” (i.e. chlorophyll/cell). This should be addressed too.
  6. On lines 262-263 it is written that “an actinic saturating pulse was emitted to record minimum fluorescence…”. This is strictly not true, since the saturating pulse will generate MAXIMUM fluorescence (Fm). Fo is the fluorescence value BEFORE the saturating light pulse is applied. So, actually, no saturating pulse is necessary for measuring Fo – just read the fluorescence value before the saturating pulse is given! [A: Yes, pressing the saturating light button is a convenient way for the instrument to display both Fo and Fm (and is a must for Fv/Fm measurements), but the PAM-flurometry-non-informed reader will erroneously understand from the text that the saturating light generates Fo; B: I have always avoided calling the saturating light a “pulse” because it can be mistaken for the ‘pulse’ in Pulse-Ampliture Modulated, PAM. Instead, please call it a “[for photosynthesis] short saturating light flash” or the like – just not pulse!
  7. Lines 379-382 are unclear: From where in Fig. 7 can one see the stark difference in Fv/Fm between fed (0.473) and unfed (0.043) slugs? If the authors refer to the 14-day data, please say so… since there is a continuous decrease for the unfed slugs, the downslope of which would probably continue also after that time.  

Some very minor points include a few syntax errors in the English (to mind comes line 345, which should read “effect on Fv/Fm of the sea slugs’…”). However, in all, this paper is written nicely and clearly. I think the above points can eaisly be taken into consideration in a revised version, which can then be published in Animals.  

Reviewer 2 Report

It is a broadly accepted fact that while certain sea slugs feed on prey containing symbiotic algae, and thereby become green, the sea-slug Berghia does not maintain a symbiotic relation with the algae it gains from its prey, an anemone. This study, by using a non-invasive approach of chlorophyll fluorescence, reaches similar conclusions. This paper broadens the evidence of a non-symbiotic relation, and as such it should be published. My only comment: as symbiosis is the central issue, authors in the Discussion should present some definition, and some discussion. of the term Symbiosis.

Reviewer 3 Report

Review of   

“Prevalence and photobiology of photosynthetic dinoflagellate endosymbionts in the nudibranch Berghia stephanieae”, submitted to Animals.

Overall comments:

Firstly, I would like to thank the editor and the authors for the opportunity to review this manuscript.

The authors investigated the photosynthetic efficiency of the Berghia stephanieae-Symbiodiniaceae association. They also evaluated the different effects of offering symbiotic and asymbiotic prey items. The findings are relevant, especially considering that a lot of questions persist regarding the nature of this association. It also does seem to fit the journal scope.

The writing style needs a bit more improvement (see specific comments below) and, in general, the relevant literature has been covered.

I have only two major (constructive) points of criticism: first, I believe that the passages on the trophic behaviour are rather confusing (see comments below) and need considerable reworking. Second, I feel that the discussion needs a bit more discussing. Especially regarding the nature of this association and the role that symbiont identity may play.

However, these issues can be easily addressed and I believe that, after corrections, this manuscript will make a fine contribution.

Therefore, I recommend a moderate revision.

Best regards

Specific comments:

Title: The title is clear and objective.

Abstract

  1. Line 26: “end-up” is not exactly the best writing style.

  1. Line 28: I do not understand how the bleaching passage makes sense here; do you mean that slugs or anemones are bleaching? It’s a bit confusing.

  1. Line 29: the scenario is unclear; slugs deprived of food or anemones without microalgae?

Introduction

  1. Line 46: Currently, the Symbiodinium genus only encompasses phylotypes that were previously designated to Clade A. The appropriate terminology is now “Symbiodiniaceae dinoflagellates” – please refer to LaJeunesse et al. (2018) in Curr Biol for the current understanding of Symbiodiniaceae taxonomy. I would also avoid using the term “zooxanthellae” as it is not specific to this taxa.

  1. Line 47: I wouldn’t call it “famous”; but “well-known” or similar.

  1. Line 54: I would add Carroll & Kempf (1990) and Monteiro et al. (2019) to the bracket that contains ref #5.

  1. Lines 60-63: This is a long sentence; I would break into two sentences.

  1. Line 65: They feed on pallida as well. The way it is written, it seems that they feed solely on E. diaphana and that this is the only species within the genus.

  1. Lines 62-63: not all those are genes, and some of these contain non-coding regions, therefore I wouldn’t refer to them as genes, but “markers”.

  1. Line 65: Too many taxa; use “Cnidaria: Actiniaria”.

  1. Line 72: “researchers do this” or “researchers do that” isn’t the proper writing style. Just mention that it is considered a good model organism.

  1. Line 78: “… E. diaphana, ON which B. stephanieae predates…”

  1. Line 86: avoid writing “John and Sara et al. did this or that”

  1. Line 86: remove comma after “introduced”. In fact, this sentence is a bit long and needs to be reduced.

  1. “methodology” or “method”?

  1. No need to uppercase in “chlorophyll”.

  1. Line 96: what is a “trophic regime”?

Materials and Methods

  1. Lines 112-116: please state which primers correspond to forward and reverse.

  1. Line 123: an annealing temp of 45ºC is very low; any reasons for such?

  1. What was the amplicon size targeted for each marker?

  1. Line 134: what do you mean by two independent reactions?

  1. Line 135: it is important to at least mention the sequencing procedures. What platform was used?

  1. Line 174: I don’t think they were juveniles; 15 mm organisms are more than capable of spawning.

  1. Was there any sort of mesh screen to prevent nudibranchs from entering filters, skimmers, sumps and etc?

  1. “photosynthetic dinoflagellate endosymbionts” can be reduced to “Symbiodiniaceae dinoflagellates”.

  1. Do you know what Symbiodiniaceae phylotype was present in B. stephanieae?

  1. Line 225: now I understand what you mean by trophic regimes. But I do not believe it is accurate; a “trophic mode” would be autotrophy, heterotrophy, osmotrophy, etc. Perhaps you mean “diets”? Or do you presume that feeding on anemones with symbionts will allow nudibranchs to perform autotrophy (and thus become mixotrophic), and that feeding on bleached anemones will make them strict heterotrophs? This needs to be clear, but “trophic regime” is making matters confusing.

  1. Line 266: what was the optimization procedure?

  1. The methods in section 2.8 are very good.

  1. Line 287: But no analytical standard was used?

  1. Line 291: read odd; a word is missing perhaps?

  1. Line 303: how were data transformed? Log?

Results

  1. Lines 308-316: this does indicate that the animals were indeed B. stephanieae and E. diaphana; but it merely indicates. I recommend that a phylogenetic tree is provided to confirm. Especially considering that the taxonomy for both taxa is still shady.

  1. 2 is excellent, but requires a bit more info. For instance, there is no mention of the dinoflagellates.

  1. 3 seems a bit unnecessary; perhaps just describe the parameters in the text? But this is more of an editorial decision than mine.

  1. Line 326: “photosynthetic”.

  1. From line 338, there are too many short paragraphs. It is a bit unusual.

  1. Line 338: this is a high degree of mortality; it clearly did not influence the results, but do you have any ideas for what may have caused it?

  1. Lines 349-351: very interesting results.

  1. Line 368: “lowest value”

  1. Line 385: I wouldn’t say that bleaching was “performed”; “achieved”, perhaps?

  1. 6-8: please state the anemone species in the captions.

Discussion

  1. I believe that, although very important, the trophic passages are confusing. I also believe that this is worth discussing more; we know little about the trophic behavior of this organism.

  1. Lines 474-485: I would a bit more information on the criteria that show that this is not likely a mutualistic association.

  1. I also feel that the symbiont identity may have played a role; perhaps a different symbiont phylotype would have rendered different results? Please remember that Symbiodiniaceae is remarkably diverse, not only taxonomically but also functionally.
